# Novel Multitarget Hydroxamic Acids with a Natural Origin CAP Group against Alzheimer’s Disease: Synthesis, Docking and Biological Evaluation

**DOI:** 10.3390/pharmaceutics13111893

**Published:** 2021-11-08

**Authors:** Margarita Neganova, Yulia Aleksandrova, Evgenii Suslov, Evgenii Mozhaitsev, Aldar Munkuev, Dmitry Tsypyshev, Maria Chicheva, Artem Rogachev, Olga Sukocheva, Konstantin Volcho, Sergey Klochkov

**Affiliations:** 1Institute of Physiologically Active Compounds of the Russian Academy of Sciences, 142432 Moscow, Russia; neganovam@ipac.ac.ru (M.N.); aleksandrova@ipac.ac.ru (Y.A.); chicheva@ipac.ac.ru (M.C.); 2N.N. Vorozhtsov Novosibirsk Institute of Organic Chemistry, Siberian Branch of the Russian Academy of Sciences, 630090 Novosibirsk, Russia; suslov@nioch.nsc.ru (E.S.); mozh@nioch.nsc.ru (E.M.); amunkuev@nioch.nsc.ru (A.M.); tsypyshev@nioch.nsc.ru (D.T.); rogachev@nioch.nsc.ru (A.R.); volcho@nioch.nsc.ru (K.V.); 3Discipline of Health Sciences, College of Nursing and Health Sciences, Flinders University, Bedford Park, SA 5042, Australia; olga.sukocheva@flinders.edu.au

**Keywords:** natural compounds, hydroxamic acids, camphane, fenchane, adamantane, molecular docking, histone deacetylase 6, β-amyloid aggregation, 5xFAD transgenic mice, Alzheimer’s disease

## Abstract

Hydroxamic acids are one of the most promising and actively studied classes of chemical compounds in medicinal chemistry. In this study, we describe the directed synthesis and effects of HDAC6 inhibitors. Fragments of adamantane and natural terpenes camphane and fenchane, combined with linkers of various nature with an amide group, were used as the CAP groups. Accordingly, 11 original target compounds were developed, synthesized, and exposed to in vitro and in vivo biological evaluations, including in silico methods. In silico studies showed that all synthesized compounds were drug-like and could penetrate through the blood–brain barrier. According to the in vitro testing, hydroxamic acids **15** and **25**, which effectively inhibited HDAC6 and exhibited anti-aggregation properties against β-amyloid peptides, were chosen as the most promising substances to study their neuroprotective activities in vivo. All in vivo studies were performed using 5xFAD transgenic mice simulating Alzheimer’s disease. In these animals, the Novel Object Recognition and Morris Water Maze Test showed that the formation of hippocampus-dependent long-term episodic and spatial memory was deteriorated. Hydroxamic acid **15** restored normal memory functions to the level observed in control wild-type animals. Notably, this effect was precisely associated with the ability to restore lost cognitive functions, but not with the effect on motor and exploratory activities or on the level of anxiety in animals. Conclusively, hydroxamic acid **15** containing an adamantane fragment linked by an amide bond to a hydrocarbon linker is a possible potential multitarget agent against Alzheimer’s disease.

## 1. Introduction

For more than three decades, hydroxamic acids have been one of the most promising and actively studied classes of chemical compounds. In medicinal chemistry, hydroxamic acids are used in the development of antitumor [1], antimalarial [2], and anti-tuberculosis drugs [3] and for the treatment of cardiovascular and other diseases [4,5].

The analysis of experimental data shows the presence of a wide spectrum of biological activities for hydroxamic acids and allows us to consider them as promising candidates for combating neuropathologies (Figure 1) [6]. For instance, the application of hydroxamic acid vorinostat led to an improvement in the spatial memory of 20-month-old mice, leading simultaneously to the expression of acetylated histone proteins [7]. Similar neuroprotective activity was found for the pan-selective histone deacetylase inhibitor panobinostat [8] and for trichostatin A [9]. It has also been shown that scriptaid, trichostatin A, vorinostat, and some other compounds based on hydroxamic acid have antioxidant potential due to their ability to restore glutathione levels [10,11,12], reduce the content of malondialdehyde [13], and prevent excess production of reactive oxygen species, protecting neurons from death [14,15]. Of particular interest is the fact that a number of hydroxamic acids have shown antiaggregation properties against the pathological β-amyloid peptide, which is considered one of the main causes of neurotoxicity in Alzheimer’s disease [16]. Thus, the pretreatment of neuroblastoma cells with a natural derivative of hydroxamic acid deferoxamine reduces the neuronal toxicity caused by β-amyloid, as well as decreasing Aβ deposits and improving the cognitive functions of APP/PS1 transgenic mice [17]. The inhibition of amyloid peptide aggregation has also been shown for batimastat, which may also be one of the possible mechanisms of its neuroprotective action [18].

Despite the great structural diversity, hydroxamic-acid-based HDAC inhibitors mainly include three pharmacophore groups: a zinc-binding group, a linker, and a CAP group (Figure 2).

The most frequent and promising directions for the development of new drugs based on hydroxamic acids are the modification of the CAP group and the linker region. Such modifications can serve as one of the most effective strategies for improving the HDAC inhibition profile by enhancing the affinity of compounds with surface enzyme groups and achieving selectivity [19]. Currently, one of the most promising directions for the modification of hydroxamic acids is the combination of a fragment of a drug or a natural compound and a hydroxamate group in one molecule. This approach makes it possible to obtain selective drugs with a high affinity for HDAC [20,21].

Exploring new methods for to the treatment of Alzheimer’s disease, we attempted to develop multitarget HDAC6 inhibitors aimed at a number of linked targets using fragments of adamantane and natural terpene compounds of camphane and fenchane as a CAP group, which were combined with linkers of various nature with an amide group. This possibility has been discussed in various works [22,23] and is a promising direction in science [24,25,26].

Adamantane, which can be found in petroleum and the monoterpene compounds camphane and fenchane themselves, has demonstrated promising pharmacological effects that are potentially useful for Alzheimer’s disease (AD) treatment, such as antioxidant effects and neuroprotective properties; moreover, one of the few anti-Alzheimer drugs, memantine, has been derived on the basis of adamantane [27,28]. In addition, it has been shown that memantine-based hydroxamic acids are able to penetrate the blood–brain barrier [29]. These facts suggest that our approach is promising.

Thus, in this study, we used natural adamantane and terpene compounds camphane and fenchane, combined with an amide group linker, as a CAP group, to obtain new multitarget potential drugs for AD treatment. This complex synthesis approach was used to obtain synergistic effects without losing efficacy in relation to HDAC6. Accordingly, 11 target compounds were developed, synthesized, and evaluated in silico, in vitro, and in vivo using various biological methods and models.

## 2. Materials and Methods

### 2.1. Chemistry

All chemicals were purchased from commercial sources (Sigma Aldrich (St. Louis, MO, USA), Acros Organics (Geel, Belgium)) and used without further purification. ^1^H and ^13^C NMR spectra were recorded on a Bruker AV-300 spectrometer (Bruker Corporation, Billerica, MA, USA) (300.13 MHz and 75.46 MHz, respectively), Bruker AV-400 (Bruker Corporation, Billerica, MA, USA) (400.13 MHz and 100.61 MHz), Bruker DRX-500 (Bruker Corporation, Billerica, MA, USA) (500.13 MHz and 125.76 MHz). Mass spectra (70 eV) were recorded on a DFS Thermo Scientific high-resolution mass spectrometer. A PolAAr 3005 polarimeter (Optical Activity, Ramsey, UK) was used to measure optical rotations [α]_D_. Melting points were measured on a Mettler Toledo FP900 Thermosystem apparatus (Mettler Toledo, Cornellà de Llobregat, Spain). Merck silica gel (Merck, Darmstadt, Germany, 63−200 μm) (63−200 μm) was used for column chromatography. Spectral and analytical measurements were carried out at the Multi-Access Chemical Service Center of Siberian Branch of Russian Academy of Sciences (SB RAS).

### 2.2. General Method for the Synthesis of Hydroxamic Acids **1** and **2**

Synthesis was performed by the method described in [30], with the excess of potassium carbonate and hydroxylamine hydrochloride being increased to 4. NH_2_OH·HCl (3.3 g, 48 mmol) and K_2_CO_3_ (6.7 g, 48 mmol) were dissolved in water (40 mL) and ethyl acetate (40 mL) was added. After the resulting mixture was cooled to 0 °C, 20 mL of ethyl acetate solution of corresponding acyl chloride (2.4 g, 12 mmol) was added dropwise and the reaction mixture was stirred at room temperature overnight. Layers were separated and the aqueous layer was extracted with EtOAc (3 × 30 mL), the combined organic layer was consequently washed with water (3 × 30 mL) and brine (3 × 15 mL), and it was dried over Na_2_SO_4_. The solvent was evaporated and the product was isolated by recrystallization, with ethyl acetate used as a solvent.

### 2.3. N-Hydroxyadamantane-1-Carboxamide **1**

Off-white solid powder, mp = 149.0–149.3 °C, yield 44%. ^1^H spectrum is in agreement with the one described in [30]. ^13^C NMR (126 MHz, DMSO-d6) δ 174.15, 38.60, 36.20, 27.61. HRMS: m/z 195.1256 (M^+^ C_11_H_17_O_2_N_1_, calc. 195.1254).

### 2.4. N-Hydroxyadamantane-2-Carboxamide **2**

Yellow solid powder, mp = 173.4 °C, yield 68%. ^1^H NMR (400 MHz, DMSO-d_6_) δ 1.48 (m, 2H), 1.64–1.73 (m, 4H), 1.73–1.80 (m, 3H), 1.83 (m, 1H), 1.99–2.11 (m, 4H), 2.33 (s, 1H), 8.55 (s, 1H), 10.29 (s, 1H). ^13^C NMR (75 MHz, DMSO-d6) δ 171.22, 46.60, 38.03, 37.16, 32.69, 29.36, 27.14, 26.84 (for spectra, see Appendix A). HRMS: m/z 195.1255 (M^+^ C_11_H_17_O_2_N_1_, calc. 195.1254).

### 2.5. General Method for Hydroxamic Acid **11**–**16** Synthesis

Corresponding amine (9.6 mmol) was slowly added to a solution of suberic or azelaic anhydride (10.2 mmol) in dry THF (90 mL) at 0 °C with vigorous stirring; the resulting mixture was stirred overnight at room temperature. The solution was filtered and evaporated, giving a corresponding acid, which was used further without purification. A solution of amido acid in dry THF (90 mL) was treated with ethyl chloroformate (18.0 mmol) and triethylamine (19.5 mmol) at 0 °C and consequently stirred at room temperature for 30 min and filtered. At the same time, hydroxylamine hydrochloride (37.9 mmol) and potassium hydroxide (37.8 mmol) solutions in methanol (25 mL each) were prepared. The methanol solution resulting from their mixing with following filtration was added to the amido acid solution in THF; the reaction mixture was stirred overnight, filtered, evaporated, suspended in chloroform, filtered again, evaporated, dissolved in 300 mL of EtOAc, washed consequently with 5% NaOH and brine to isolate unreacted amido acid, dried over Na_2_SO_4_, evaporated, and purified by column chromatography, with CHCl_3_/MeOH 0–5% mixture used as an eluent. Yields of hydroxamic acids were calculated based on the starting amines.

### 2.6. N^1^-Hydroxy-N^8^-((1R,2R,4R)-1,7,7-Trimethylbicyclo [2.2.1]heptan-2-yl)octanediamide **11**

Off-white solid powder, mp = 94.8 °C, yield 30%. ^1^H NMR (300 MHz, DMSO-d_6_) δ 0.73 (s, 3H), 0.76 (s, 3H), 0.88 (s, 3H), 1.05–1.18 (m, 2H), 1.19–1.26 (m, 4H), 1.39–1.51 (m, 5H), 1.56–1.67 (m, 4H), 1.92 (t, *J* = 7.4 Hz, 2H), 1.99–2.16 (m, *J* = 7.5, 2H), 3.70 (q, *J* = 7.6 Hz, 1H), 7.10 (d, *J* = 7.9 Hz, 1H), 8.63 (s, 1H), 10.31 (s, 1H). ^13^C NMR (101 MHz, DMSO-d_6_) δ 11.55, 19.96, 20.42, 25.11, 25.54, 26.77, 28.44, 32.28, 35.30, 35.72, 37.03, 44.25, 46.40, 48.68, 55.86, 169.12, 171.98. HRMS: m/z 324.2403 (M^+^ C_18_H_32_O_3_N_2_^+^, calc. 324.2407), ${\left[\alpha \right]}_{D}^{21}$, ${\left[\alpha \right]}_{D}^{21}$ = −20 (c 0.74 g/100 mL in MeOH).

### 2.7. N^1^-Hydroxy-N^9^-((1R,2R,4R)-1,7,7-Trimethylbicyclo [2.2.1]heptan-2-yl)nonanediamide **12**

Off-white solid powder, mp = 50.4 °C, yield 23%. ^1^H NMR (400 MHz, DMSO-*d*_6_) δ 0.72 (s, 3H), 0.76 (s, 3H), 0.88 (s, 3H), 1.04–1.15 (m, 2H), 1.16–1.28 (m, 6H), 1.40–1.52 (m, 5H), 1.54–1.66 (m, 4H), 1.92 (t, *J* = 7.4 Hz, 2H), 1.98–2.17 (m, 2H), 3.70 (q, *J* = 7.6 Hz, 1H), 7.13 (d, *J* = 7.9 Hz, 1H), 8.66 (s, 1H), 10.33 (s, 1H). ^13^C NMR (126 MHz, DMSO-*d*_6_) δ 11.62, 20.01, 20.48, 25.18, 25.64, 26.82, 28.59, 28.60, 28.62, 32.31, 35.33, 35.75, 37.05, 44.28, 46.46, 48.74, 55.90, 169.16, 172.05. HRMS: m/z 338.2568 (M^+^ C_19_H_34_O_3_N_2_, calc. 338.2564), ${\left[\alpha \right]}_{D}^{23}$ = −17 (c 0.64 g/100 mL in MeOH).

### 2.8. N^1^-Hydroxy-N^8^-((1R,2R,4S)-1,3,3-Trimethylbicyclo[2.2.1]heptan-2-yl)octanediamide **13**

Off-white solid powder, mp = 45.1 °C, yield 25%. ^1^H NMR (500 MHz, DMSO-*d*_6_) δ 0.70 (s, 3H), 0.93 (s, 3H), 0.94–1.02 (m, 1H), 0.99 (s, 3H), 1.10–1.15 (m, 1H), 1.18–1.26 (m, 4H), 1.31–1.41 (m, 1H), 1.41–1.50 (m,4H), 1.52–1.59 (m, 2H), 1.60–1.67 (m, 2H), 1.91 (t, *J* = 7.4 Hz, 2H), 2.12–2.17 (m, 2H), 3.48 (dd, *J* = 9.5, 1.9 Hz, 1H), 7.20 (d, *J* = 9.4 Hz, 1H), 8.69 (s, 1H), 10.36 (s, 1H). ^13^C NMR (126 MHz, DMSO-*d*_6_) δ 19.70, 21.31, 25.20, 25.66, 25.74, 26.59, 28.47, 28.52, 30.97, 32.33, 35.32, 38.63, 42.32, 47.69, 47.77, 62.85, 169.15, 173.11. HRMS: m/z 324.2404 (M^+^ C_18_H_32_O_3_N_2_, calc. 324.2407), ${\left[\alpha \right]}_{D}^{23}$ = +23 (c 0.643 g/100 mL in MeOH).

### 2.9. N^1^-Hydroxy-N^9^-((1R,2R,4S)-1,3,3-Trimethylbicyclo[2.2.1]heptan-2-yl)nonanediamide **14**

Off-white solid powder, mp = 44.9 °C, yield 22%. ^1^H NMR (500 MHz, DMSO-*d*_6_) δ 0.70 (s, 3H), 0.92 (s, 3H), 0.94–1.01 (m, 1H), 0.99 (s, 3H), 1.10–1.16 (m, 1H), 1.16–1.29 (m, 6H), 1.32–1.41 (m, 1H), 1.42–1.51 (m, 4H), 1.52–1.58 (m, 2H), 1.61–1.67 (m, 2H), 1.92 (t, *J* = 7.4 Hz, 2H), 2.15 (m, 2H), 3.48 (dd, *J* = 9.5, 1.9 Hz, 1H), 7.19 (d, *J* = 9.4 Hz, 1H), 8.69 (s, 1H), 10.34 (s, 1H). ^13^C NMR (151 MHz, DMSO-*d*_6_) δ 19.64, 21.25, 25.12, 25.61, 25.71, 26.56, 28.53, 28.57, 28.60, 30.92, 32.27, 35.28, 38.59, 42.29, 47.67, 47.72, 62.84, 169.11, 173.07. HRMS: m/z 338.2568 (M^+^ C_19_H_34_O_3_N_2_, calc. 338.2564), ${\left[\alpha \right]}_{D}^{23}$ = +23 (c 0.41 g/100 mL in MeOH).

### 2.10. N^1^-(Adamant-2-yl)-N^8^-Hydroxyoctanediamide **15**

White solid powder, mp = 133.6 °C, yield: 34%. ^1^H NMR (400 MHz, DMSO-*d*_6_) δ 1.12–1.25 (m, 4H), 1.46 (d, *J* = 11.4 Hz, 6H), 1.67 (s, 2H), 1.71 (d, *J* = 11.9 Hz, 1H), 1.75 (s, 6H), 1.80 (s, 1H), 1.87–2.00 (m, 4H), 2.12 (t, *J* = 7.4 Hz, 2H), 3.81 (d, *J* = 7.4 Hz, 1H), 7.63 (d, *J* = 7.5 Hz, 1H), 8.66 (s, 1H), 10.34 (s, 1H). ^13^C NMR (101 MHz, DMSO-*d*_6_) δ 24.7, 25.5, 28.7, 28.8, 29.4, 34.0, 36.3, 37.6, 41.6, 51.43, 51.79, 172.15, 174.24. HRMS: m/z 322.2257 (M^+^ C_18_H_30_N_2_O_3_, calc. 322.2251).

### 2.11. N^1^-(Adamantan-2-yl)-N^9^-Hydroxynonanediamide **16**

White solid powder, mp = 133.2 °C, yield: 42%.^1^H NMR (300 MHz, DMSO-*d*_6_) δ 1.23 (s, 6H), 1.41–1.52 (m, 6H), 1.62–1.86 (m, 10H), 1.86–2.04 (m, 4H), 2.12 (t, *J* = 7.3 Hz, 2H), 3.82 (d, *J* = 7.4 Hz, 1H), 7.62 (d, *J* = 7.6 Hz, 1H), 8.66 (s, 1H), 10.32 (s, 1H). ^13^C NMR (126 MHz, DMSO-*d*_6_) δ 25.14, 25.59, 26.80, 28.57, 30.99, 31.56, 32.28, 35.28, 36.91, 37.22, 52.76, 169.13, 171.69. HRMS: m/z 336.2407 (M^+^ C_19_H_32_N_2_O_3_, calc. 336.2405).

### 2.12. Methyl 4-Formylbenzoate **18**

Thionyl chloride (1.5 mL) was added to a mixture of 4-formylbenzoic acid (1 g, 6.67 mmol) in methanol (15 mL, 20 mmol) and cooled to 0 °C. The mixture was stirred overnight; then, the solvent was evaporated and HCl aqueous solution was added. The mixture was stirred for 6 h; the solid formed was filtered off, washed with water, and dried. The product yield was 0.93 g (85%). The NMR spectra were consistent with previously reported data [31].

### 2.13. (E)-3-(4-Methoxycarbonylphenyl)prop-2-Enoic Acid **19**

A solution consisting of methyl 4-formylbenzoate (0.5 g, 3.05 mmol), malonic acid (0.475 g, 4.58 mmol), and piperidine (0.25 mL) in pyridine (2.5 mL) was refluxed for 2 h. After being cooled to room temperature, the reaction mixture was poured into 1M HCl (30 mL). The precipitate was filtered, washed with water and acetonitrile, and dried to give 0.57 g (91%) of (E)-3-(4-methoxycarbonylphenyl)prop-2-enoic acid. NMR data were in agreement with [32].

### 2.14. General Procedure for the Synthesis of Esters **20**–**22**

T3P (propanephosphonic acid anhydride, 50 wt.% solution in ethyl acetate, 10 mmol) was added to a mixture of (E)-3-(4-methoxycarbonylphenyl)prop-2-enoic acid (4.85 mmol), amine (5.3 mmol), and pyridine (1.33 mL) in ethyl acetate (2.7 mL). The mixture was stirred overnight at room temperature and then water was added. The precipitate formed was filtered, washed with water, and dried. The products were used in the next step without further purification.

### 2.15. Methyl 4-((E)-3-Oxo-3-(((1S,2R,4R)-1,7,7-Trimethylbicyclo[2.2.1]heptan-2-yl)amino)prop-1-en-1-yl)benzoate **20**

Off-white solid powder, mp = 78.5 °C, yield: 85%. ^1^H NMR (400 MHz, CDCl_3_) δ 0.88 (d, *J* = 11.3 Hz, 6H), 0.98 (s, 3H), 1.20 (ddd, *J* = 13.3, 9.2, 4.1 Hz, 1H), 1.36 (ddd, *J* = 12.7, 9.3, 3.7 Hz, 1H), 1.58–1.83 (m, 4H), 1.94 (dd, *J* = 13.3, 9.1 Hz, 1H), 3.93 (s, 3H), 4.07 (td, *J* = 9.0, 5.0 Hz, 1H), 5.60 (d, *J* = 9.0 Hz, 1H), 6.46 (d, *J* = 15.5 Hz, 1H), 7.57 (d, *J* = 8.1 Hz, 2H), 7.64 (d, *J* = 15.5 Hz, 1H), 8.04 (d, *J* = 8.1 Hz, 2H). ^13^C NMR (126 MHz, CDCl_3_) δ 11.58, 20.10, 20.21, 26.84, 35.83, 38.92, 44.78, 46.98, 48.72, 52.00, 56.95, 76.64, 76.90, 77.15, 123.46, 127, 42, 129.82, 130.52, 139.19, 139.29, 164.59, 166.40. HRMS: m/z 341.1983 (M^+^ C_21_H_27_N_1_O_3_, calc. 341.1986), ${\left[\alpha \right]}_{D}^{23.5}$ = −58 (c 0.92 g/100 mL in CHCl_3_).

### 2.16. Methyl 4-((E)-3-Oxo-3-(((1S,2R,4S)-1,3,3-Trimethylbicyclo[2.2.1]heptan-2-yl)amino)prop-1-en-1-yl)benzoate **21**

Off-white solid powder, mp = 94.7 °C, yield: 70%. ^1^H NMR (300 MHz, CDCl_3_) δ 0.80 (s, 3H), 1.06 (s, 3H), 1.14 (s, 3H), 1.17–1.35 (m, 3H), 1.45 (ddd, *J* = 12.8, 8.0, 4.9 Hz, 1H), 1.59–1.89 (m, 3H), 3.77 (dd, *J* = 9.6, 1.7 Hz, 1H), 3.90 (s, 2H), 5.60 (d, *J* = 9.5 Hz, 1H), 6.53 (d, *J* = 15.6 Hz, 1H), 7.55 (d, *J* = 8.3 Hz, 2H), 7.63 (d, *J* = 15.6 Hz, 1H), 7.97–8.07 (m, 2H). ^13^C NMR (126 MHz, CDCl_3_) δ 19.45, 21.05, 25.77, 27.16, 30.73, 39.28, 42.53, 48.01, 48.40, 52.03, 63.59, 76.65, 76.90, 77.15, 123.28, 127.46, 129.85, 130.56, 139.22, 139.40, 165.87, 166.44. HRMS: m/z 341.1981 (M^+^ C_21_H_27_N_1_O_3_, calc. 341.1986), ${\left[\alpha \right]}_{D}^{23.5}$ = +11 (c 0.96 g/100 mL in CHCl_3_).

### 2.17. Methyl 4-((E)-3-((Adamantan-2-yl)amino)-3-Oxoprop-1-en-1-yl)benzoate **22**

White solid powder, mp = 170.9 °C, yield: 88%. ^1^H NMR (400 MHz, CDCl_3_) δ 1.51–1.83 (m, 12H), 1.95–2.01 (m, 2H), 3.90 (s, 3H), 4.14–4.22 (m, 1H), 5.95 (d, *J* = 8.0 Hz, 1H), 6.50 (d, *J* = 15.5 Hz, 1H), 7.55 (d, *J* = 8.3 Hz, 2H), 7.62 (d, *J* = 15.6 Hz, 1H), 7.98–8.04 (m, 2H). ^13^C NMR (151 MHz, CDCl_3_) δ 26.93, 27.06, 31.76, 31.80, 36.94, 37.33, 52.06, 53.50, 123.60, 127.42, 129.85, 130.49, 139.20, 164.38, 166.44. HRMS: m/z 339.1826 (M^+^ C_21_H_25_N_1_O_3_, calc. 339.1829).

### 2.18. General Procedure for the Synthesis of Target Hydroxamic Acids **23**–**25**

KOH (11.71 g, 209.1 mmol) was added to a mixture of NH_2_OH·HCl (9.8 g, 141 mmol) in MeOH (50 mL). The mixture was stirred for 30 min, followed by filtering the precipitate. The filtrate was added to a solution of ester (4.1 mmol) in MeOH (10 mL) and the solution was stirred overnight at room temperature. The solvent was evaporated under reduced pressure and water was added to the reaction mixture. The solution was neutralized by HCl conc. till pH 6–7 was reached. The precipitated solid was collected, washed with water, and dried. The product was purified by recrystallization from CHCl_3_, unless otherwise specified.

### 2.19. N-Hydroxy-4-((E)-3-Oxo-3-(((2R,4R)-1,7,7-Trimethylbicyclo[2.2.1]heptan-2-yl)amino)prop-1-en-1-yl)benzamide **23**

White solid powder, mp = 202.5 °C, yield: 57%. ^1^H NMR (400 MHz, DMSO-*d*_6_) δ 0.78 (d, *J* = 4.6 Hz, 6H), 0.94 (s, 3H), 1.03–1.24 (m, 2H), 1.47–1.58 (m, 1H), 1.58–1.76 (m, 4H), 3.85 (td, *J* = 8.5, 5.4 Hz, 1H), 6.90 (d, *J* = 15.8 Hz, 1H), 7.41 (d, *J* = 15.9 Hz, 1H), 7.54 (d, *J* = 8.1 Hz, 1H), 7.62 (d, *J* = 8.1 Hz, 2H), 7.78 (d, *J* = 8.0 Hz, 2H), 9.10 (s, 1H), 11.28 (s, 1H). ^13^C NMR (126 MHz, DMSO-*d*_6_) δ 11.89, 20.30, 20.59, 26.99, 36.02, 37.61, 44.52, 46.79, 49.24, 56.53, 124.62, 127.71, 133.31, 137.62, 138.02, 163.89, 164.71. HRMS: m/z 342.1943 (M^+^ C_20_H_26_N_2_O_3_, calc. 342.1938), ${\left[\alpha \right]}_{D}^{26.5}$ = −66 (c 0.66 g/100 mL in MeOH).

### 2.20. N-Hydroxy-4-((E)-3-Oxo-3-(((1S,4S)-1,3,3-Trimethylbicyclo[2.2.1]heptan-2-yl)amino)prop-1-en-1-yl)benzamide **24**

Off-white solid powder, 159.9–161.0 °C, yield: 47%. ^1^H NMR (400 MHz, DMSO-*d*_6_) δ 0.74 (s, 3H), 0.98 (s, 3H), 1.06 (s, 3H), 1.18 (d, *J* = 9.7 Hz, 1H), 1.42 (s, 1H), 1.54–1.76 (m, 5H), 3.63 (dd, *J* = 9.4, 1.9 Hz, 1H), 7.01 (d, *J* = 15.8 Hz, 1H), 7.42 (d, *J* = 15.7 Hz, 1H), 7.61 (dd, *J* = 14.5, 8.8 Hz, 3H), 7.79 (d, *J* = 8.0 Hz, 2H), 9.09 (s, 1H), 11.28 (s, 1H). ^13^C NMR (101 MHz, DMSO-*d*_6_) δ 19.70, 21.30, 25.64, 26.63, 30.93, 42.34, 47.72, 48.03, 63.40, 124.26, 127.40, 127.52, 133.10, 137.43, 137.86, 163.70, 165.55. HRMS: m/z 342.1933 (M^+^ C_20_H_26_N_2_O_3_, calc. 342.1938), ${\left[\alpha \right]}_{D}^{23.5}$ = +4 (c 0.92 g/100 mL in MeOH).

### 2.21. 4-((E)-3-((-Adamantan-2-yl)amino)-3-Oxoprop-1-en-1-yl)-N-Hydroxybenzamide **25**

Off-white solid powder, 222.5–222.6 °C, yield: 50% (purification by silica column chromatography, eluent—chloroform/methanol). ^1^H NMR (300 MHz, DMSO-*d*_6_) δ 1.53 (d, *J* = 12.6 Hz, 2H), 1.64–1.93 (d, *J* = 3.3 Hz, 10H), 2.01 (d, *J* = 12.8 Hz, 2H), 3.99 (d, *J* = 7.4 Hz, 1H), 6.98 (d, *J* = 15.8 Hz, 1H), 7.43 (d, *J* = 15.7 Hz, 1H), 7.62 (d, *J* = 8.1 Hz, 2H), 7.79 (d, *J* = 8.0 Hz, 2H), 8.01 (d, *J* = 7.8 Hz, 1H), 9.11 (s, 1H), 11.26 (s, 1H). ^13^C NMR (126 MHz, DMSO-*d*_6_) δ 26.79, 26.74, 31.05, 31.65, 36.85, 37.15, 53.07, 124.48, 127.36, 127.47, 133.12, 137.31, 137.80, 163.62, 164.04. HRMS: m/z 340.1777 (M^+^ C_20_H_24_N_2_O_3_, calc. 340.1781). 

#### 2.21.1. In Vitro

##### Lipid Peroxidation

To obtain the brain homogenate, rats preliminarily anesthetized with CO_2_ were decapitated with a guillotine. The brain was homogenized in a buffer containing KCl (120 mM) and HEPES (20 mM), pH = 7.4 at 4 °C, and centrifuged at 1500 rpm to obtain a supernatant. The protein was quantified according to the standard microbiuret method [33].

To investigate the effect of the compounds on lipid peroxidation (LPO) of rat brain homogenate, we used the modified version of the TBARS test [34], which is based on the reaction of 2-thiobarbituric acid with the end LPO product—malonic dialdehyde (MDA). The plate well, according to the experimental scheme, contained the compounds under study (100 μM), rat brain homogenate (2 mg/mL), as well as ferrous iron ions (FeSO_4_·10H_2_O) at a concentration of 500 μM and 1.6 mM *tert*-butyl hydroxyperoxide as an initiator. After 30 min of incubation at 37 °C, TBARS reagent was added to all samples, incubated for 90 min at 90 °C, and centrifuged at 6000 rpm for 15 min. The optical density of the selected supernatant was measured on a Victor 3 plate reader (Perkin Elmer) at λ = 540 nm.

##### Antiradical Activity

The DPPH test was used to examine antiradical activity [35] based on the ability of a stable chromogen radical of 2,2-diphenyl-1-picrylhydrazyl (DPPH) to remove a hydrogen atom, which is accompanied by a change in the absorption maximum in the electronic absorption spectra. The amount of reduced DPPH was measured using a Cytation ™ 3 multifunctional plate analyzer (BioTek Instruments, Inc., Winooski, VT, USA) at a wavelength of 517 nm.

##### HDAC6 Inhibiting Properties

The HDAC6 enzyme activity was determined by the degree of substrate deacetylation with the fluorescence method based on the study of the substrate deacetylation kinetics in the presence of the enzyme, using a commercially available kit (Enzo Life Sciences—FLUOR DE LYS^®^ HDAC6 fluorometric drug discovery assay kit, Enzo Biochem, New York, NY, USA). All procedures were carried out in accordance with the attached protocol. Fluorescence was measured on an EnVision spot fluorometer at λ_ex_ = 350–380 nm, λ_em_ = 440–460 nm.

##### Docking

For all calculations, data from the Protein Data Bank [36] and modules of the software package [37] were used for further processing with the ‘LigPrep’ module, and geometric optimization was performed in OPLS3e [38]. The search for possible tautomeric and ionized conditions at pH 7.0 +/− 2.0 was carried out using ‘Epik’ [39,40]. A detailed description of the docking procedure is given in the Appendix A.

##### Antiaggregation Activity

The effect of the studied compounds on the aggregation of β-amyloid (1–40/1–42) was analyzed using the method of recording Thioflavin T fluorescence (10 µM) [41] for 72 h at 37 °C. The fluorescence signal was measured using a multifunctional flatbed analyzer, the Cytation ™ 3 (BioTek Instruments, Inc., Winooski, VT, USA), at λ_ex_ = 450 nm, λ_em_ = 480 nm.

##### Influence on Cell Viability

The effect of the hydroxamic acids on the viability of cells obtained from human embryonic kidneys, HEK 293, as well as the native form of human neuroblastoma, SH-SY5Y, was evaluated in the MTT test [42]. Using a multifunctional flatbed analyzer, Cytation™3 (BioTek Instruments, Inc., Winooski, VT, USA), the optical density was determined at a wavelength of 530 nm. Based on the dose-dependent curves, the concentration values leading to 50% inhibition of cell population growth (IC_50_) were calculated.

#### 2.21.2. In silico Evaluation of Pharmacokinetic Parameters of Synthesized Compounds

Pharmacokinetic parameters (ADME/Tox) were determined using the ‘QikProp’ subroutine [37]. The methodology for predicting molecular properties is based on the Monte Carlo method. The molecules were compared with the properties of known drugs: the prediction was based on the generation of a ‘similarity matrix’ of fragments of the studied molecule. A full description of the in silico evaluation of the pharmacokinetic parameters of the synthesized compounds is given in the Appendix A.

#### 2.21.3. Plasma Sample Preparation and LC–MS/MS Analysis

EDTA-stabilized rat blood plasma was used in the study. The spikes were prepared by adding 100 µL of the solution of a compound to 900 µL of plasma, followed by continuous shaking. A 100 µL aliquot of a sample was taken and mixed with 900 µL of MeOH, shaken for 15 min, and centrifuged. Then, 100 µLc of the supernatant was transferred to a vial insert and analyzed.

HPLC–MS/MS analyses were performed using a Shimadzu LC-20AD Prominence chromatograph equipped with a SIL-20AC cooled autosampler and a gradient pump. A column filled with a reversed-phase ProntoSil 120-AQ C18 sorbent (Econova, Novosibirsk) was used. Water with the addition of 0.1% HCOOH was used as mobile phase A, and methanol containing 0.1% HCOOH was used as mobile phase B. The gradient was as follows: 0 min—5% B; 1 min—5% B; 2 min—95% B; 5 min—100% B; the flow rate was 300 μL/min; the injection volume was 10 μL. The column was then equilibrated for the following analysis.

Mass spectrometric detection was carried out on a 3200 QTRAP mass spectrometer (SCIEX, USA) using electrospray ionization. The following conditions were used for the analysis: positive MRM mode, CUR (curtain gas) = 30 psi, CAD (collision-activated dissociation gas) = High, IS (ion source voltage) = 5500V, TEM (temperature) = 350 °C, GS1 (sprayer gas) = 20 psi, GS2 (evaporator gas) = 20 psi, dwell time = 100 msec. The detection parameters of the compounds **15** and **25** in MRM mode are given in Table 1. The instrument was controlled, and information was collected using the Analyst 1.6.3 software (AB SCIEX, Singapor); chromatograms were processed using the MultiQuant 2.1 software (AB SCIEX, Singapor).

#### 2.21.4. In Vivo

##### Animals and Experimental Groups

All animal procedures were approved by the Animal Care and Use Committee of IPAC RAS. For in vivo studies, 11-month-old male mice of the 5xFAD line (Tg(APPSwFlLon,PSEN1*M146L*L286V) 6799Vas/J) were used. The animals were housed in cages with free access to water and food in a room with a 12 h/12 h light–dark cycle, at a temperature of 23 ± 1 °C and humidity of 50 ± 5%. Before the series of experiments, animals were randomly divided into four groups (*n* = 8 per group): (1) wild-type group (C57BL6/j mouse line); (2) 5xFAD group; (3) 5xFAD + **15** group; and (4) 5xFAD + **25** group.

Both compounds **15** and **25** were dissolved in saline solution and DMSO (10%) immediately before use and intraperitoneally administered to animals for 20 days at a dose of 15 mg/kg body weight for experiments. In the control group, the vehicle solution (saline and DMSO (10%), intraperitoneal) was administered with the same time schedule.

##### The Open Field Test

The open field box consisted of a square, non-transparent, grey box (40 cm × 40 cm × 40 cm). Illumination intensity was kept at 50 lux. During a session, a single animal was allowed to explore the empty open field, and its baseline level of activity was measured. The session lasted for 10 min (Figure 5A). The apparatus was cleaned thoroughly with 70% ethanol and dried after each session to remove scent cues. The entire test was videotaped and processed with a computer operated in the EthoVision XT system (Noldus, Wageningen, Netherlands). The number of standing events was manually counted in a blind manner by one observer. The parameters to be considered were the number of standing events, distance traveled in the arena, and mean rate of movement, to study the explorative and locomotor components; and time spent in the central and peripheral areas, to evaluate anxiety levels [43].

##### The Novel Object Recognition Test

The Novel Object Recognition Test assesses a mouse’s ability to remember if it has previously encountered an object or not. The same box that had been used in the open field test was used for the Novel Object Recognition Test. Mice were habituated to the box one day before the test, during the open field test. On the familiarization day, mice were placed into the box for the acquisition period with two objects (A and B) and were allowed to explore and familiarize themselves with the objects for 10 min. The mice were given a 24 h inter-trial interval and then placed back in the test box. On the test day, all conditions were the same as during the acquisition period, except that one of the two objects was replaced with a novel object (C). During the testing period, mice were allowed to explore both of the objects for 10 min (Figure 6A). All trials were videotaped and analyzed with a computer-operated EthoVision XT system (Noldus, Wageningen, the Netherlands). An interaction was considered when the mouse’s nose touched the object or was pointed towards the object within a 1 cm radius. Successful memory formation in mice was indicated if a mouse spent more time investigating and exploring an object that it had never seen (novel object) than an object that it had encountered before (familiar object) [44].

##### Morris Water Maze Test

The Morris Water Maze Test was performed based on the Morris paradigm, 1985, modified [45,46]. This test allowed us to evaluate spatial learning and memory formation in mice. The experimental apparatus consisted of a circular pool (diameter, 150 cm; height, 60 cm; OpenScience, Russia) containing water at 22 ± 1 °C. The lighting conditions were as follows: dark side of the pool was 50 lx; bright side was 75 lx. The pool was equally divided into quadrants. A platform (diameter, 10 cm) was positioned inside the reservoir, placed 1 cm below the water surface in the target quadrant of the maze. The platform and pool were white with anti-glare coating, which, together with shadowless lamp lighting, created the effect of an invisible platform that was hidden under the water. As visual cues, we used 4 figures (OpenScience, Moscow, Russia) fixed on the racks and located at the pool sides.

The test is based on two phases: the acquisition phase (training days) and the retention phase (probe trial) (Figure 7A). Initially, the training session was performed, during which each animal was placed into the water facing the pool wall. After placing, each animal was given 60 s to find and mount onto the hidden platform. If it failed to locate the platform during the allocated time, it was guided gently to swim to the platform and allowed to stay on it for 30 s. Each mouse received training sessions, and the test was repeated four times for four consecutive days in acquisition training (sum 16 trials for training). All 4 start positions for each day (north, south, east, west) were deliberately randomized. The probe trial was performed on the 5th day after the training phase: there were no platform and the mice were given 90 s for swimming. The criteria for the effectiveness of learning and memory formation were the time spent in the target and wrong quadrants, the latency period, and the number of entries to the platform area.

#### 2.21.5. Ex Vivo

At the end of the in vivo series of experiments, the animals’ brains were sampled for further measurement of the lipid peroxidation level, as well as glutathione content in mouse brain homogenates. A synaptosomal p2 fraction containing mitochondria was also isolated to investigate the bioenergetic characteristics of organelles.

##### MDA Level and Glutathione Content

The intensity of lipid peroxidation of the mouse brain homogenate was determined using the TBA test, similar to the method described above, but without initiators.

The performance of the glutathione system of the cell’s own antioxidant defense was evaluated using a commercially available kit (Glutathione Assay Kit, Sigma Aldrich, Saint Louis, MO, USA) in accordance with the protocol provided by the manufacturer. The total glutathione level was determined by spectrophotometric measurement at a wavelength of 412 nm by the amount of 2-nitro-5-thiobenzoic acid, a product formed during the reduction of GSH 5,5′-dithiobis-(2-nitrobenzoic acid).

##### Bioenergetic Characteristics of the Mitochondria

The study on the effect of the electron transport chain complexes was carried out on a preparation of the mitochondrial p2 fraction of the brain (10 µg/well) using the Agilent Seahorse XF96e Analyzer (Seahorse Bioscience, Santa Clara, CA, USA) to measure the rate of oxygen uptake under the action of modulators. For this purpose, the activators of the electron transport chain complex I-glutamate/malate, the inhibitor of complex I, rotenone (2 µM), the substrate of complex II, potassium succinate (2 µM), the inhibitor of complex III, antimycin A, and the substrates of complex IV, ascorbate, were used/TMPD (0.5 µM).

##### Histology and Histochemistry

For histological examination, the mouse brain was fixed in 10% neutral buffered formalin (Leica Biosystems Inc., Buffalo Grove, IL, USA) at +4 °C for 24 h. Then, the material was dehydrated, defatted, and clarified in a Leica ASP200 apparatus (Leica Biosystems Inc., Buffalo Grove, IL, USA) using ethanol-xylene wiring according to the following scheme: deionized water—2 h; 70% ethanol—12 h; ethanol 96% sequentially—1 h, 30 min, 2 portions 15 min each; a mixture of ethanol and xylene 1:1—20 min; xylene—2 portions for 30 min each; paraffin—3 times for 1 h each. Then, the material was embedded in paraffin blocks using a Leica EG1160 apparatus (Leica Biosystems Inc., Buffalo Grove, IL, USA). Then, the paraffin blocks were cut using a Leica RM 2265 rotary microtome (Leica Buffalo Grove, IL, USA), section thickness 8 μm, and mounted on Leica X-tra Adhesive glasses (Leica Biosystems Inc., Buffalo Grove, IL, USA) with poly-lysine. Dewaxing was performed using a Leica ST 5020 robot (Leica Biosystems Inc., Buffalo Grove, IL, USA) according to the following scheme: 3 portions of xylene for 5 min each, 2 portions of 96% ethanol for 15 min each, 75% ethanol for 10 min, then distilled water for 5 min [47,48]. After this, sections were subjected to histochemical staining to visualize β-amyloid aggregates, using the following scheme: Congo red (Sigma Aldrich, Saint Louis, MO, USA)—25 min, tap water—5 min, Mayer’s hemalum (Potassium hydroxide (Sigma Aldrich, Saint Louis, MO, USA) in 80% ethanol 100 mL)—5 min, tap water—10 min [49]. Then, the sections were embedded under coverslips using Immu-Mount (Thermo Shandon Ltd., Runcorn, Halton, UK). The results were evaluated using a ZEISS LSM 880 laser scanning microscope with an Airyscan module (Carl Zeiss Vision, Jena, Germany). Shooting was carried out at the same settings, using the ‘tile scan’ function of the microscope. Image processing was performed using ImageJ 1.52 (1.8.0_172). A 10 × 10 grid was imposed on images containing only brain tissue; then, in 10 grid cells located obliquely from top left to right bottom corner [50], we selected plaques using the ‘magic wand’ tool, so the average plaque area and luminosity were calculated. Three images were analyzed for each animal in each group; then, the average of three animals was calculated.

### 2.22. Data Analysis

During the in vivo experiments, none of the animals were excluded. The data were expressed as mean ± SEM, and statistical comparisons were made using analysis of variance (ANOVA) followed by Bonferroni post hoc tests. Two-way repeated measures (mixed model) ANOVA followed by Bonferroni post hoc tests were used to compare the two objects in the object recognition task. The data analysis for the ex vivo experiments was performed using one-way analysis of variance (ANOVA) followed by Dunnett’s multiple comparison tests. *p* ≤ 0.05 was considered statistically significant. The statistical analysis was performed using GraphPad Prism 5 (GraphPad Software, San Diego, CA, USA).

## 3. Results and Discussion

### 3.1. Chemical Synthesis

Hydroxamic acids **1** and **2** were synthesized starting from corresponding acid chlorides. As 2-adamantane carboxylic acid chloride **3** is not commercially available, it was synthesized in a four-step sequence (Figure 1) starting from adamantan-2-one 4 according to [51].

Hydroxamic acids **1** and **2** were obtained using the method described in [30], with the excess of potassium carbonate and hydroxylamine hydrochloride being increased to 4 (Figure 2). The 1H-NMR spectrum of N-hydroxyadamantane-1-carboxamide is in good agreement with that presented in the literature [30].

During further steps in hydroxamic acid synthesis, bornylamine **5** and fenchylamine **6** were obtained in accordance with the method described previously [52]. Corresponding oximes **7** and **8** were prepared by refluxing (+)-camphor or (−)-fenchone with hydroxylamine, followed by reduction with a NiCl_2_/NaBH_4_ system, leading to a mixture of *exo*-/*endo*-diastereoisomers with a 6/1 ratio for bornylamine and 2/3 for fenchylamine. The major *exo*-stereoisomer **5** and *endo*-stereoisomer **6** were isolated by column chromatography, with CHCl_3_ being used as the eluent (Figure 3).

Suberic and azelaic anhydrides **9** and **10** were synthesized with the method described in [53,54]. By their interaction with amines **5**, **6**, and commercially available 2-aminoadamantane in dry THF, followed by treatment with ethylchloroformate and hydroxylamine, hydroxamic acids **11**–**16** were obtained (Figure 4). Yields of derivatives **11**–**16** were calculated based on the starting amines.

To obtain hydroxamic acids with the cynnamyl core, 4-formylbenzoic acid **17** was converted into corresponding methyl ester **18** followed by the Knoevenagel condensation–decarboxylation process to form carbocyclic acid **19** with a 77% overall yield, as described in [31,32]. The reaction of amines **5**, **6**, and 2-aminoadamantane with compound **19** in the presence of a mild coupling reagent, T3P, led to the formation of amides **20**–**22** with good yields. Further interaction of the compounds with hydroxylamine in MeOH resulted in the formation of target compounds **23**–**25** (Figure 5).

### 3.2. Biological Evaluation

Neurodegenerative diseases, including Alzheimer’s disease (AD), are polyetiological and often diagnosed late, when many neurological changes are irreversible and pharmacological corrections are ineffective [55,56]. Due to the complexity of AD characteristics, the search for substances that have multi-targeting action on some important pathogenesis chains is the main approach to the development of effective drugs. The required drug should not only target symptomatic components and reduce cognitive impairment but also radically prevent the development of a number of pathological neurodegenerative processes. Oxidative stress and misfolded proteins, and disturbances in epigenetic regulation, such as overexpression and aberrant activity of HDAC6, are some of the interactive targets involved in the determination of the abnormal neuroenvironment and AD-linked impairments.

In this study, we used in vitro, in silico, in vivo, and ex vivo methods to investigate the neuroprotective activity of the synthesized hydroxamic acids with a natural origin CAP group. We also examined the processes associated with oxidative stress and mitochondrial dysfunction in post-mortem brain samples We determined the antioxidant status of compounds, the effect on HDAC6 activity, the formation of β-amyloids, and the cell survival. We analyzed the ADME/Tox profile, as well as assessing the action of the generated substances on the cognitive functions of transgenic 5xFAD animals. The observed effects were discussed in association with oxidative stress and mitochondrial dysfunction characteristics detected in post-mortem brain samples.

#### 3.2.1. In Vitro

##### Antioxidant Activity

The main target of free radicals, the overproduction of which is observed in Alzheimer’s disease, is lipids of both cell membranes and organelle membranes. Therefore, important properties for potential neuroprotectors are both the presence of direct antiradical activity to neutralize reactive oxygen species (ROS) and the ability to inhibit the process of lipid peroxidation. To verify the test, the known antioxidant Trolox was used as a reference compound.

The antiradical activity of the test compounds was evaluated by the DPPH test, based on the ability of antioxidants to donate an electron to the stable chromogen radical DPPH, converting it into a nonradical form and neutralizing its activity. As shown in Table 2, hydroxamic acid **25** had the most pronounced antiradical properties, as evidenced by the percentage of activity at the level of 43.34%. The chromogenic free radical DPPH used in this test has a large size, and substances even with a similar chemical structure may have different antiradical properties due to spatially difficult access. It is likely that compound **25** had a high affinity for this free radical.

A rat brain homogenate preparation containing fragments of cell membranes was used as a model system to investigate the antioxidant activity of the hydroxamic acids. To initiate the process of lipid peroxidation, ferrous ions were used to trigger the Fenton reaction, during which the iron valence changes and a highly reactive hydroxyl radical is formed, and *tert*-butyl hydroxyperoxide as a direct source of free radicals. Most hydroxamic acids exhibit moderate antioxidant activity in this test, while, for compounds **15** and **25**, the most pronounced ability to inhibit *t*-BHP-induced LPO was observed—by 43.24% and 44.59%, respectively.

Apparently, the presence of antioxidant properties for the synthesized hydroxamic acids can be associated with the inclusion of adamantane, fenchane, and camphane fragments in the CAP group. The presence of antioxidant activity for a number of adamantane derivatives is known from the literature [27]. Thus, for the adamantane derivative memantine, the literature describes the ability to reduce oxidative damage in the cortex and hippocampus of the rat brain, two important brain regions involved in the formation of memory [57].

##### Inhibition of HDAC6

The ability to modulate the activity of histone deacetylases is the main criterion that determines the prospects of hydroxamic acid compounds as therapeutic agents [58,59,60]. In Alzheimer’s disease, increased activity and overexpression of histone deacetylase 6 is recorded, which is accompanied by disturbances in the regulation of transcription and correlates with the accumulation of β-amyloids, hyperphosphorylation of tau protein, and neuronal degeneration [61]. In this regard, we investigated the effect of the synthesized compounds on the HDAC6 activity. The study was carried out with the method based on monitoring the deacetylation of a substrate, which is an acetylated lysine side chain, followed by the formation of a fluorophore as a result of treatment with a developer reagent. The known HDAC inhibitor, Trichostatin A, was used as a positive control.

For most of the investigated hydroxamic acids with a natural origin CAP group, inhibitory ability towards HDAC6 was found (Table 2). Exceptions were compounds **1** and **2**, which lacked the linker part in the structure, and the CAP group was directly related to the hydroxamate function. Probably, due to this, there is no penetration of the zinc-binding fragment of the substance into the catalytic center of the enzyme [58] and its inhibition. It was shown that compounds **11**, **15**, and **16**, with a linear hydrocarbon linker structure containing the framework fragments of adamantane and camphor in the CAP group, exhibited the highest inhibitory ability towards HDAC6. The IC_50_ values of the HDAC inhibitory effect by these hydroxamic acids were in the nanomolar range: 0.69 µM, 0.96 µM, and 0.74 µM, respectively. Moreover, high activity was observed for compounds **12**, **14**, **24**, and **25**, as evidenced by the IC_50_, not exceeding 6.5 µM.

##### Docking

An in silico approach was used to determine possible interaction modes of synthesized hydroxamic acids **2, 11**–**16,** and **23**–**25** containing a naturally occurring CAP group, with the active center of HDAC6. The crystal structure with the zinc-containing binding site ID 5EDU of HDAC6 domain 2 (reference ligand Trichostatin A [62]) was extracted from RSCD PDB [https://www.rcsb.org, accessed date 10 May 2021] for the molecular docking procedure. The ‘Glide’ extra-precision protocol of docking was used at the first step of modeling [63,64,65]; the best poses obtained as results of this procedure then were used for the following MM_GBSA calculations, in order to obtain the ΔG_bind_ values. The value of RMSD of reference Trichostatin A obtained as a re-docking result was 1.5030 Å (see Appendix A).

The calculated data for compounds **2**, **11**–**16**, and **23**–**25** (number and types of interactions with the key amino acids (AAs) residues of the 54EDU binding site) are summarized in Table 3. Figure 3A–C show specific examples of the docking poses of the most active compounds, **11**, **15,** and **16**, in the HDAC6 catalytic domain. A detailed description of the results obtained is given in the Appendix A. It was shown that all tested hydroxamic acids could be divided into several conventional groups according to the features of their interaction with the HDAC6 catalytic center. For compounds **23** and **25**, with a cynnamyl linker present the first group, an interaction with the 5EDU binding site of HDAC6 via hydrogen bonds with GLY619 was found. The second group of compounds was represented by ligands **11**, **12**, **13**, **14**, **15**, and **16** with a hydrocarbon (C_6_-C_7_) linker, which were grouped according to the mode of interaction with the amino acid residues of the binding site. They all formed hydrogen bonds with HIS610, HIS651, and TYR782 and quite similar (with small deviations) hydrophobic interactions with PRO608, PHE620, PHE679, PHE680, LEU749, and TYR782. The π-π-stacking interaction was registered for compound **24** (overlap with PHE620) and for compound **25** (with PHE680).

Thus, it can be assumed that the nature of the linker greatly contributes to the HDAC6-inhibiting ability of the synthesized hydroxamic acids. Obviously, the best action was found for compounds **11**, **12**, **13**, **14**, **15**, and **16** with a hexa- and heptamethylene linker, which have a structure corresponding to the classical pharmacophore model of HDAC6 inhibitors—‘CAP-linker-ZBD’.

In order to analyze the correlation between the experimental data of the synthesized compounds’ HDAC6 inhibitory activity (IC50) and in-silico-predicted ability to bind to the 5EDU site of HDAC6 (*Ki*), the values of *Ki* were calculated according to the following formula: $$K_{i}=e^{-\frac{\Delta G_{bind}}{RT}}$$
where
$$∆G_{\mathrm{bind}}\;-\;(\mathrm{cal}/\mathrm{mol});\;T\;=\;298\;K;\;R\;=\;1.987\;\frac{\mathrm{cal}}{\mathrm{mol}\times K}$$

Primary data about *G_bind_* values obtained as a result of the docking procedures performed, as well as calculated values of Ki, pKi, and pIC_50_, are summarized in Appendix A (see Appendix A). The correlation graph linking the pKi (obtained from docking) and pIC_50_ values (experimental) for compounds **2**, **11**–**16**, and **23**–**25** is represented in Figure 4.

It was found that three compounds with a sterically hindered amide group (**13**, **14**, and **24**) fell out of the correlation. Apparently, the presence of such steric difficulties leads to a decrease in the availability of these groups, and the compounds **13**, **14**, and **24** could interact (with comparable affinity) with other parts of HDAC6, reducing the inhibitory activity in comparison with the predicted one. Based on this, the results for compounds **13**, **14**, and **24** were excluded from the correlation analysis (Figure 4). For the remaining compounds, a good correlation coefficient of 0.88 was observed, which confirmed our assumption about the preferred interaction of these substances with the HDAC6 catalytic domain.

##### Antiaggregation Activity

A number of neurodegenerative diseases, different in their clinical picture, have a similar molecular mechanism of pathogenesis, which is based on pathological protein aggregation leading to the development of proteinopathy [66,67]. The nerve tissues of patients with Alzheimer’s disease contain protein aggregates—amyloid plaques formed by an aggregating β-amyloid peptide consisting of **40**–**42** amino acid residues.

The effect of hydroxamic acids on the stage of aggregation of the pathological β-amyloid peptide was investigated for 72 h using two model systems—fragments Aβ_1–40_ and Aβ_1–42_. The fluorophore Thioflavin T was used as an indicator of the ongoing fibrillation process, the interaction of which with proteins in the state of amyloid fibrils leads to an increase in the fluorescence signal [68].

Figure 5 shows that Thioflavin T fluorescence, indicative of the active formation of amyloid beta fibrils, increases rapidly during the first six hours and then gradually progresses during the 72 h incubation period. At the same time, pretreatment of β-amyloid preparations 1–40 and 1–42 with substances **15**, **16**, **23**, **24**, and **25** leads to a noticeable fibrillation inhibition of these peptides during incubation. All compounds containing an adamantane fragment and a linker part exhibited antiaggregation properties, with the highest activity shown for **15** and **25**. Hydroxamic acid **24** had a comparable effect. For these substances, the concentration dependences of the antiaggregation effect were also studied (Figure 5B–D, F–H). The color change in the graph reflects the incubation time of the hydroxamic acids with Aβ. With an increase in concentration, the effectiveness of the antiaggregatory action increased rapidly.

This action of hydroxamic acids can be associated with the influence of both the CAP group and the hydroxamate function, since, earlier, a similar effect was found for both adamantane derivatives [69] and for compounds based on hydroxamic acid [16,70].

##### Influence on Cell Viability

Due to the fact that most compounds of the class of hydroxamic acids have high cytotoxic activity and are mainly used as anticancer drugs [1,71], their use as neuroprotective drugs is limited. Since the goal of our work was to search for agents for the pharmacological correction of neurodegenerative diseases, a necessary part of the biological activity study of the synthesized hydroxamic acids was to determine their effect on the survival of cell cultures. In our work, we used neuron-like neuroblastoma cells SH-SY5Y and a healthy cell line isolated from the human embryonic kidney, HEK 293.

As can be seen from Table 4, all investigated hydroxamic acids did not exhibit a pronounced cytotoxic effect in relation to SH-SY5Y and HEK-293 cells, as evidenced by IC_50_ values of more than 50 µM. It should be noted that the HDAC-inhibiting activity of a number of the synthesized hydroxamic acids was in the nanomolar range, and it is obvious that the therapeutic concentration of substances will be much lower than the IC_50_ values.

#### 3.2.2. In silico Assessment of the Pharmacokinetic Parameters of the Synthesized Compounds

An important element of drug development is ADME/Tox studies, which help to determine the prospects of a candidate drug and describe the following: substance bioavailability, distribution in the body, metabolism, elimination time, and potential toxicity. An important property for a potential neuroprotective drug is the ability of substances to penetrate into the blood–brain barrier; therefore, prior to the in vivo experiments, an in silico analysis of the pharmacokinetic parameters of the hydroxamic acids under study was carried out. A detailed description of the results obtained is given in the Appendix A.

As the analysis of the pharmacokinetic (ADME/Tox) parameters of the synthesized compounds showed, all compounds could be considered potential drugs (drug-like) with a high degree of permeability across the blood–brain barrier (predicted brain/blood partition coefficient).

#### 3.2.3. Stability Study of the Compounds **15** and **25** in Blood Plasma of Rats

To study the stability of the compounds, their solutions (spikes) in rat blood plasma were prepared with a concentration of 1000 ng/mL. For this, 100 μL of a solution (10 mcg/mL) of each compound in methanol was added to 900 μL of plasma. After preparation, the spike was placed on a shaker and shaken throughout the experiment. Aliquots of the spike with a volume of 100 μL were taken after 5, 15, 30, 60, 120, 180, and 270 min, after which the sample was mixed with MeOH, shaken for 10 min, then centrifuged and analyzed with LC–MS/MS.

The compounds were detected in positive MRM mode using ESI. Optimization of parameters for the analysis of the compounds was performed by infusion of solutions of both substances (1000 ng/mL) in a mixture of water–methanol (20:80, *v*/*v*) containing 0.1% formic acid, using a syringe pump integrated into the mass spectrometer. The compositions of the mobile phase and elution gradient were developed in order to reduce peak tailing and achieve better sensitivity of the analysis. Methanol containing 0.1% of formic acid was found to provide better peak intensity than ACN with the same modifier.

It was found that when compound **15** was introduced into rat plasma, it degraded quite rapidly. Figure 6A shows the time dependence of the peak area of the compound in chromatograms. The half-life of the substance, estimated from the graph in the figure, was 60–70 min. The results obtained indicate that, apparently, there were enzymes in the plasma that destroyed the substance.

In contrast, compound **25** spiked in rat blood plasma was found to be stable at least for 4.5 h (Figure 6B). Presumably, its stability was caused by conjugation of the hydroxamic moiety with the aromatic ring, leading to delocalization of the electron pairs of nitrogen and oxygen, reducing the reactivity of the functional group. This was indirectly confirmed by the fact that the peak area of compound **25** on the chromatogram was significantly lower when compared with that of compound **15** when both substances were analyzed at the same concentration. This indicated the much less basic properties of compound **25** in its protonation in the electrospray, which was due to the presence of nitrogen atoms.

#### 3.2.4. In Vivo

The main goal of our work was to identify leading compounds from a number of synthesized hydroxamic acids with a natural origin CAP group, exhibiting a complex effect on several chains in the pathogenesis of Alzheimer’s disease, such as processes associated with oxidative stress, the activity of the HDAC6 enzyme, and aggregation of the pathological β-amyloid peptide. According to the results of the in vitro testing, hydroxamic acids **15** and **25** were selected as the most promising substances for the study of neuroprotective activity in vivo. They effectively inhibited HDAC6 and, at the same time, exhibited antiaggregation properties against β-amyloid peptides. These substances also had antioxidant activity and did not show cytotoxic effects in relation to non-cancer HEK 293 cell culture.

The initial stage of the in vivo investigation was the study of the toxic effects of the leading compounds in clinically healthy animals—C57Bl6/j male mice. (IPAC RAS provided full approval for this research (Approval No 61 date 18/04/2021)). The substances were administered intraperitoneally at different concentrations (maximum 300 mg/kg). No pathological changes in the behavior or physiological condition of the animals were observed. The injection of an equivalent volume of solvent (10% DMSO in saline) also did not lead to any pathological changes. Thus, the results enable experiments on animals at the selected effective concentration.

The in vivo neuroprotective potential of hydroxamic acids **15** and **25** was assessed by their effect on the general behavior and cognitive functions of 5xFAD transgenic male mice (Tg(APPSwFlLon,PSEN1*M146L*L286V) 6799Vas/J) [72] at the age of 11 months. The pathological phenotype of this mouse line includes amyloid deposits, gliosis, neurodegeneration, memory impairment, intracellular Aβ accumulation, and pronounced neuronal death, found in Alzheimer’s disease. As a control group, we used wild-type animals of the same age—C57Bl6/j male mice.

The effect on the general behavioral characteristics of the animals was investigated in the open field test, which evaluated exploratory and motor activity, as well as the level of anxiety. The cognitive-stimulating effect of the hydroxamic acids was determined by their effect on episodic memory in the New Object Recognition Test and spatial memory in the Morris Water Maze Test.

##### Open Field Test

In order to determine the effects of hydroxamic acids **15** and **25** on the general behavior in mice, we first evaluated explorative and locomotor activity and also anxiety levels in an open field test on 5xFAD mice and their wild-type littermates. In this test, **15** and **25** did not show any alteration in the number of standing events, distance traveled in the arena, or mean movement rate, as well as the time spent in the central and peripheral areas by transgenic animals. Interestingly, there was a tendency toward a smaller number of standing events in 5xFAD mice, but not in treated animals, compared with WT animals (*p* = 0.077, Figure 7B).

Thus, no differences in general behavioral characteristics were found between 5xFAD and transgenic mice treated by **15** and **25** compounds compared with WT. 

##### Novel Object Recognition

Object recognition memory is a type of declarative memory that crucially depends on the function of the hippocampus. We performed a behavioral object recognition task to measure episodic memory. The object recognition tasks performed here require associative processing to encode the relationships between various elements encountered during a given exposure [73]. Compared to 5xFAD, both wild-type and 5xFAD animals treated with compound **15** showed an increased preference for the novel object in the testing trial (Figure 8B). While, for WT mice, the difference in this parameter was not significant, 5xFAD mice treated with hydroxamic acid **15** significantly showed a strong preference and spent more time studying a new object compared to the previously studied one (*p* = 0.033). At the same time, untreated transgenic animals or mice that were treated with hydroxamic acid **25** began to show memory impairment when presented with a new object. This demonstrated problems with associative learning and hippocampus-dependent episodic memory formation.

Thus, it is likely that the impairment in the object recognition task could have resulted from a deficit in recollection, a process underlying recognition memory [74].

###### Morris Water Maze Test

Learning Phase:

The Morris Water Maze Test was used to test the effect of hydroxamic acids **15** and **25** on learning and long-term spatial memory formation. The escape latency (time required to reach the platform) was used to evaluate learning capacity. During the learning trials, the mice in all groups except for the 5xFAD showed a significant improvement in latency time to find the submerged platform (Figure 9B). Thus, the mean latency time to reach the hidden goal among untreated transgenic mice decreased only to 50.47 ± 4.68 s (*p* = 0.85) on the last training day. In turn, both for wild-type animals and for mice treated with hydroxamic acids **15** and **25**, the difference between the first and fourth training days was significant and decreased from 53.33 ± 3.68 s to 34.58 ± 4.19 s (*p* = 0.003), from 53.76 ± 2.79 s to 34.28 ± 4.32 s (*p* = 0.001), and from 53.46 ± 2.75 s to 38.63 ± 3.01 s (*p* = 0.016), respectively. Moreover, a significant decrease in this parameter in **15**-treated mice was recorded already on the third training day (*p* = 0.031). These results suggest that compounds **15** and **25** had the ability to recover the learning impairments of 5xFAD animals.

Another important approach when studying animals’ behavior in the Morris Water Maze is the analysis of the animals’ strategy in finding a hidden platform during the entire training period. The analysis was first introduced by Wolfer et al., 2001 [75]. It lies in the gradual switch from hippocampus-independent egocentric navigation to hippocampus-dependent allocentric navigation when animals search for the platform during the training for several days. 

Hippocampus-independent egocentric navigation includes several types of target-seeking strategies (Figure 9C): (1) thigmotaxis—the animal moves only along the periphery of the arena; (2) random search—the animal begins to move away from the periphery of the arena with visible movements inward; (3) scanning—behavior associated with random search, focused on the center of the swimming pool. This behavior is often observed in the first days when an animal is placed in the experimental arena, and the mouse demonstrates a stereotypical sequence of patterns when searching for the hidden platform, focusing only on proximal and internal signals. Then, as learning progresses, the hippocampus-dependent allocentric orientation begins to work with the addition of various motion vectors using distal signals (visual cues located at the sides of the pool). These strategies (Figure 9C) include: (1) directed search—swimming with small circular or winding movements to find a platform; (2) focal search—this behavior is also associated with random searches, but, here, an animal is actively looking for a specific small section of the arena; (3) direct swimming—an animal goes directly to the platform. Thus, the behavior of an animal in the Morris Water Maze depends on both egocentric and allocentric search strategies, and the contribution of the latter increases when placing animals from different starting points of the route and repeating tests, which characterizes the success of hippocampus-dependent spatial learning.

When analyzing the strategies used by the animals to search for the platform, it was found that untreated 5xFAD transgenic mice preserved the advantage of using hippocampus-independent egocentric navigation up to the fourth training day, in contrast to wild-type animals. This confirms the known data that the pathological phenotype of this animal line is characterized by disorders in the neuron functioning in the hippocampal region and indicates the absence of effective learning. In turn, hydroxamic acid **15** led to the normalization of this situation and the successful formation of the allocentric cognitive map of the like in the control group. In the **25**-treated group, there was a similar trend but to a lesser extent (Figure 9D).

Probe Trail:

Furthermore, we studied spatial memory formation using the probe test. The memory formation was evaluated according to several indicators, including the latency period, the number of entries to the platform area (the previous platform location) (Figure 10B,C), as well as time spent in the target (the platform-located quadrant during the acquisition trials) and wrong quadrants (Figure 10D,E). As shown in Figure 10A–C, hydroxamic acid **15** intervention significantly reduced the latency for entering the platform zone (*p* = 0.013) and increased entries to the platform (*p* = 0.042) by these AD mice up to the level of control wild-type animals. In turn, an improvement in these spatial memory parameters was also observed for mice pretreated with hydroxamic acid **25**, but significant differences were found in the time spent in the wrong quadrant compared to native 5xFAD animals (*p* = 0.013) (Figure 10E).

Thus, hydroxamic acids **15** and **25** have been proven to improve spatial memory and learning ability in transgenic mice 5xFAD modeling Alzheimer disease in the Morris Water Maze Test.

#### 3.2.5. Ex Vivo

As mentioned earlier, oxidative stress and mitochondrial dysfunction are important pathogenetic components of AD [56,76,77]. These pathological characteristics have also been demonstrated using various AD mouse models [78,79,80].

After the in vivo experiments, brain samples were taken from all animals to determine indicators of oxidative stress within, such as the MDA level, showing the intensity of lipid peroxidation, as well as the total pool of glutathione, as an endogenous antioxidant system marker.

The glutathione content was estimated using the method based on measurement of the absorption of a colored compound formed during the colorimetric reaction of DTNB with SH groups of glutathione. It was found that there were pathological changes in the glutathione redox system in transgenic mice, which appeared as a significant decrease in total glutathione to 7.034 ± 0.602 mmol/mg protein at 11.100 ± 0.593 mmol/mg protein in wild-type animals (*p* < 0.0001, Figure 11A). At the same time, mice pretreated with hydroxamic acid **15** tended to recover this parameter (*p* = 0.086 versus 5xFAD mice). Obviously, the described disorders of the glutathione chain of the antioxidant system in 5xFAD mice can cause an imbalance in the functioning of the entire antioxidant system—in particular, the activation of lipid peroxidation processes, which is considered the main molecular mechanism involved in oxidative damage to cellular structures [81]. Therefore, the level of lipid peroxidation of mouse brain homogenates was also measured by its final product, malondialdehyde. It was shown that transgenic animals are characterized by an intensification of this process, as evidenced by an increase in the MDA level (*p* < 0.05 versus WT mice). No effect was found for the tested compounds (Figure 11B).

To evaluate mitochondrial functioning, we determined the functioning of the respiratory chain complexes using Seahorse Bioanalyzer. The analysis of the mitochondrial respiratory conditions according to the mitochondrial electron flow demonstrated that transgenic 5xFAD animals had lower oxygen consumption than wild-type mice after injections of succinate and ascorbate/TMPD. However, the oxygen consumption rate (OCR) was increased in transgenic animals treated with the test compounds. Thus, after injection of succinate, **15** increased the OCR by organelles of transgenic mice from 119.07 ± 25.51 pmol/min to 251.00 ± 10.45 pmol/min, and after the injection of ascorbate, from 308.78 ± 59.54 pmol/min to 422.67 ± 14.61 pmol/min (Figure 12A). At the same time, for compound **25**, a similar effect was shown with respect to the stimulation of the IV complex of the mitochondrial respiratory chain (Figure 12B).

Thus, when mitochondrial ETC complex function was evaluated, significant alterations were observed in complex II and complex IV, demonstrating that the ETC was functionally different in 5xFAD mice compared with WT animals. Based on the results on the studied substances, it can be strongly suggested that the mitochondria have a greater ability to respond to insults, as denoted by the increased oxygen consumption rate.

Although the AD etiology is not fully understood, convincing evidence has accumulated for a correlation between accumulation Aβ and cognitive impairments [82]. To characterize the Aβ plaques, we stained mice brain sections with Congo red fluorescent dye, which detects the amyloid core of mature plaques. It was established that positive staining was detected in all groups of 11-month-old transgenic mice, but not in wild-type animals (Figure 13A). Additionally, our quantification revealed a significantly lower Congo red-positive amyloid burden in mice treated with hydroxamic acid **15**. This compound led to a remarkable reduction of 45% in plaque load (*p* = 0.0338) compared to the untreated 5XFAD mice (Figure 13B). No changes in β-amyloid plaque level were found for animals treated with compound **25**.

## 4. Conclusions

In this study, we performed directed synthesis and analyzed the effects of HDAC6 inhibitors. The synthesized agents contained fragments of adamantane and natural terpenes camphane and fenchane combined with linkers of various nature with an amide group, which were used as the CAP group. A total of 11 target compounds were developed, synthesized, and extensively investigated using a complex of in vitro, in vivo, and ex vivo assays. Using in silico studies, we determined that all synthesized compounds were drug-like and could penetrate through the blood–brain barrier. According to the results of the in vitro testing, hydroxamic acids **15** and **25**, which effectively inhibited HDAC6 and exhibited antiaggregation properties against β-amyloid peptides, were chosen as the most promising substances for the study of neuroprotective activity in vivo. These substances also demonstrated antioxidant activity without cytotoxic effects in healthy HEK 293 cell culture.

All in vivo studies were performed using 5xFAD transgenic mice simulating Alzheimer’s disease. In these animals, the Novel Object Recognition and Morris Water Maze Test showed that the formation of hippocampus-dependent long-term episodic and spatial memory was deteriorated. Hydroxamic acid **15** restored normal cognitive functions in the AD mice models to the level observed in control wild-type animals. Notably, this effect was precisely associated with the ability to restore lost cognitive functions without changes in motor and exploratory activities or the level of anxiety in animals. For compound **25**, similar effects were observed, but to a lesser extent.

Thus, the obtained results support the prospective therapeutic use of our synthetic agents as effective polyfunctional neuroprotective compounds. Data indicate that hydroxamic acid **15**, containing an adamantane fragment linked by an amide bond to a hydrocarbon linker, is a possible potential multitarget agent against Alzheimer’s disease.

## Data Availability

The following are available online at Appendix A containing docking and NMR spectra for compounds.

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
