# Peer review of "Novel Multitarget Hydroxamic Acids with a Natural Origin CAP Group against Alzheimer’s Disease: Synthesis, Docking and Biological Evaluation"

_pharmaceutics, 2021, doi:10.3390/pharmaceutics13111893_

Round 1

Reviewer 1 Report

The authors report novel derivatives for the treatment of Alzheimer's disease. The manuscript is well written and the experiments are adeguate. All compounds were chemically characterized. Minor point: are the final compound stable? If the authors immagine an oral administration of the compounds, they should evaluate the stability in gastro-intestinal fluids and plasma. These data could improve the quality of this paper.

Author Response

Answers to Reviewer 1

We are grateful to the respected reviewer for carefully reading our article and for the valuable comments made.

Below are the answers to the comments of the respected reviewer.

Minor point: are the final compound stable? If the authors immagine an oral administration of the compounds, they should evaluate the stability in gastro-intestinal fluids and plasma. These data could improve the quality of this paper.

Experiments on the stability of the most active compounds will be carried out at the stage of pharmaceutical development. If necessary, we plan to chemically modify the compounds to form a stable dosage form. When planning in vivo experiments at this stage, we took into account several factors that could affect the therapeutic efficacy of the synthesized compounds. Therefore, the choice of the route of administration of hydroxamic acids to transgenic mice was primarily due to the use of dimethyl sulfoxide (10%) as a co-solvent for complete and stable dissolution of substances, administered for similar purposes by the intraperitoneal route. In addition, since the target organ for the action of our compounds is the brain: for better and faster penetration of hydroxamic acids into the bloodstream and passage through the blood-brain barrier, intraperitoneal administration is preferable to oral administration.

Reviewer 2 Report

The manuscript pharmaceutics-1437532 "Novel multitarget hydroxamic acids with a natural origin CAP group against Alzheimer's disease: synthesis, docking and biological evaluation" by Margarita Neganova et al describe the synthesis, characterization and biological activity of some derivatives of hydroxamic acid.  The synthesis of new compounds was confirmed by 1H, 13C NMR, and HRMS. The antioxidant, cytotoxic and anti-aggregation activity has also been evaluated.
The manuscript is well-written and presents a great interest to the readers of Pharmaceutics.

As comments/suggestions: 

  1. Regarding the statement presented in lines 185-186, how can explain the more pronounced antiradical properties for compound 25 compared to the other compounds?
  2. What are the free radicals produced by Fenton-type reactions (rows 189-190) that are neutralized by the synthesized compounds?

  3. Rows 190-193: how can explain the difference of antioxidant activity in the series of the synthesized compounds?
  4. The saline solution of synthesized compounds with DMSO 10% used for intraperitoneally administered to animals isn't too toxic?

Author Response

Regarding the statement presented in lines 185-186, how can explain the more pronounced antiradical properties for compound 25 compared to the other compounds?

Since the chromogenic free radical DPPH used in the technique has a large size and a specific chemical structure, substances even with a similar chemical structure may have different antiradical properties due to spatially difficult access. Probably, compound 25 has a high affinity for this free radical.

What are the free radicals produced by Fenton-type reactions (rows 189-190) that are neutralized by the synthesized compounds?

In our work to determine the antioxidant activity of the tested compounds, we used the TBA test, where in a model system - a rat brain homogenate - with the help of iron ions, the process of lipid peroxidation was triggered by means of the Fenton reaction, during which the iron valence changes and a highly reactive hydroxyl radical is formed. We assume that substances that have shown an inhibitory effect on LPO have the ability to bind free radicals in this system, in particular, the hydroxyl radical, which is also confirmed by the data obtained in the study of antiradical activity using the DPPH test.

Rows 190-193: how can explain the difference of antioxidant activity in the series of the synthesized compounds?

Apparently, the observed differences in the antioxidant properties of the synthesized hydroxamic acids can be associated with their differences in the chemical structure, in particular, the fragments of the CAP group.

The saline solution of synthesized compounds with DMSO 10% used for intraperitoneally administered to animals isn't too toxic?

The use of such a concentration of DMSO, a universal and widely used as a solvent, is acceptable, since a number of studies have confirmed its obvious low toxicity when used at concentrations ≤ 10% [Galvao J, Davis B, Tilley M, Normando E, Duchen MR, Cordeiro MF. Unexpected low-dose toxicity of the universal solvent DMSO. FASEB J. 2014 Mar;28(3):1317-30. doi: 10.1096/fj.13-235440; Kais, B., Schneider, K., Keiter, S., Henn, K., Ackermann, C., Braunbeck, T., 2013. DMSO mod-ifies the permeability of the zebrafish (Danio rerio) chorion-implications for the fishembryo test (FET). Aquat. Toxicol. 140, 229–238]. So, in the work [ModrzyÅ„ski JJ, Christensen JH, Brandt KK. Evaluation of dimethyl sulfoxide (DMSO) as a co-solvent for toxicity testing of hydrophobic organic compounds. Ecotoxicology. 2019 Nov; 28 (9): 1136-1141. doi: 10.1007/s10646-019-02107-0] DMSO concentrations up to 10% had no effect on cell survival, and in the study [Cavas M, Beltrán D, Navarro JF. Behavioural effects of dimethyl sulfoxide (DMSO): changes in sleep architecture in rats. Toxicol Lett. 2005 Jul 4;157(3):221-32. doi: 10.1016/j.toxlet.2005.02.003] 5 и 10% DMSO did not induce neurotoxic effects or behavioral changes in rats, which also highlights its suitability as a co-solvent to dissolve hydrophobic chemicals, allowing them to pass through biological membranes and thus ensuring proper bioavailability of the substances. In addition, with the introduction of such a dose of DMSO in saline, no pathological reactions were observed.

English language and style are fine/minor spell check required

We have carefully checked the text of the article. The English language and style have been corrected in some cases.
